# PRRSV RNA Detection and Predictive Values Between Different Sow and Neonatal Litter Sample Types

**DOI:** 10.3390/vetsci12020150

**Published:** 2025-02-10

**Authors:** Peng Li, Isadora Machado, Thomas Petznick, Emily Pratt, Jinnan Xiao, Chris Sievers, Paul Yeske, Swami Jayaraman, Daniel C. A. Moraes, Guilherme Cezar, Mafalda Mil-Homens, Hao Tong, Kelly Will, Darwin Reicks, Jason Kelly, Onyekachukwu H. Osemeke, Gustavo S. Silva, Daniel C. L. Linhares

**Affiliations:** 1College of Veterinary Medicine, Iowa State University, Ames, IA 50010, USA; lipeng@iastate.edu (P.L.);; 2ArkCare, Omaha, NE 68132, USA; 3Swine Vet Center, Saint Peter, MN 56082, USA; 4Reicks Veterinary Research & Consulting, Saint Peter, MN 56082, USA; 5Suidae Health and Production, Algona, IA 50511, USA

**Keywords:** PRRSV detection, sample type, predictive value, TOSc, TF, sow, litter status, vertical transmission

## Abstract

Common sample types, including processing fluid, serum, and family oral fluid, can neither determine whether PRRSV infection originates vertically or horizontally nor directly reflects the sow’s PRRSV status. This study aimed to describe PRRSV RNA detection by RT-rtPCR among different sample types taken from sows (pre- and post-farrowing) and their respective litters within 12 h post-farrowing and investigated predictive values of PRRSV status of sow represented by tonsil oral scrubbing (TOSc) and that of dead piglets represented by tongue fluid (TF) to reflect the PRRSV status of respective neonatal “live litter”. The results showed that pre-farrowing TOSc samples had significantly higher PRRSV positivity than TF, serum and blood swab pools, while dead piglet serum had significantly lower mean Ct values than all other sample types. TOSc samples had 25% positive predictive value (PPV) for “live litter” PRRSV status while the pre-farrow and post-farrow TOSc had 87.2% and 89.0% negative predictive value (NPV), respectively. In conclusion, we characterized PRRSV RNA detection among all sub-populations within a litter with easy-to-use TOSc samples and neonatal litter samples, suggesting the occurrence of vertical transmission 90 days post-LVI in sows. TOSc samples from sows had low PPV and high NPV for their respective litter’s PRRSV status.

## 1. Introduction

One major problem for Porcine reproductive and respiratory syndrome virus (PRRSV) management in the breeding herd is the viral persistence at the population level, reflecting in long time-to-stability. The American Association of Swine Veterinarians (AASV) defines that a breeding herd reaches PRRSV stability when there is diagnostic evidence of a sustained lack of viremia in pigs at weaning for 13 consecutive weeks [1]. A study in 2021 reported a medium time to stability (TTS) of 35 weeks, ranging from 23 to 49 weeks [2]. It is not uncommon for producers to report a rebreak of the same PRRSV strain in the process of virus elimination [3]. This indicates PRRSV can circulate in the herd at low prevalence for several months. This variation in time to stability can be attributed to two modes of virus transmission to the piglets: (i) vertical transmission, i.e., in uterus infection; (ii) horizontal transmission, characterized by virus transmission by pigs (either live or dead), people, tools and other contaminated vectors to naïve piglets in the farrowing after their birth [4].

There are several well-established sample types to monitor PRRSV activity in suckling pigs, including processing fluid (PF), serum, and family oral fluids (FOF) [5,6,7]. However, there are two specific questions that cannot be answered by diagnostic results from these sample types. First, whether the detected virus RNA originated from vertical transmission or horizontal transmission cannot be distinguished because viremia can develop as early as within 12 h post-infection [8]. When those samples were PCR-positive, it is possible that the piglets were either infected in the uterus, by other infected pigs, or by contaminated fomites after birth [4]. To determine whether vertical transmission occurs, it is important to investigate PRRSV status in different subpopulations within a litter separately within 12 h post-farrowing to reduce the likelihood of horizontal transmission after the piglets are born.

Second, these samples target suckling piglets and may not reflect their respective dam’s PRRSV status. Sows are a major source of PRRSV infection to piglets [4]. Investigating the dynamics of PRRSV detection from different sample types of both sows and piglets is required to provide a holistic picture of whole-herd viral distribution. Moreover, as sow samples can be collected before piglets are born, estimating predictive values of PRRSV status in sow samples for their respective litters can be of great value to implement strategic interventions such as test-and-removal or test-and-segregation of positive sows [9].

Conventional sow sampling tools such as serum and tonsil scraping are time-consuming, labor-intensive, and animal invasive because they require specialized tools and trained personnel in addition to snaring sows [10,11,12]. Recently Peng et al. reported a novel sow sampling tool, tonsil oral scrubbing (TOSc), which was adapted from a sow collector reported for the test-and-removal of African swine fever virus-infected sows in China [13]. TOSc takes biological samples consisting of fluids from the oral and tonsillar area of a sampled sow within seconds without snaring the sows and shows comparable positivity with tonsil scraping [13,14,15].

Postmortem tongue tissue fluid (TF) was recently described as an easy and feasible sample type for PRRSV detection and monitoring in dead animals, including stillborn and liveborn dead animals [16,17]. Since collecting blood samples from a large number of live neonatal piglets to determine their PRRSV status is challenging, investigating the predictive value of TF for assessing the PRRSV status of their respective live littermates would be worthwhile.

Thus, this was a field study monitoring individual sows before and after farrowing, and their offspring for PRRSV RNA by RT-rtPCR using different sample types. The objectives of this study are twofold:

Describe the PRRSV RT-rtPCR results (positivity and Ct values) for each set of sows and piglet specimens within 12 h post farrowing;

Investigate the positive and negative predictive values (PPV and NPV) for neonatal live piglets’ PRRSV status (positive or negative) based on RT-rtPCR results from sow TOSc and stillborn TF samples.

## 2. Materials and Methods

### 2.1. Study Design Overview

This was a field study monitoring individual sows pre- and post-farrowing, and their offspring for PRRSV RNA by RT-rtPCR in different sample types. At 90 days after whole-herd exposure to live-virus inoculation (LVI), TOSc samples were collected from 555 sows 2 weeks pre-farrowing from two breeding herds and tested for PRRSV RNA by RT-rtPCR. All RT-rtPCR-positive sows were matched with RT-rtPCR-negative sows by parity and followed until 12 h post-farrowing, when TOSc from each sow; blood swabs from each live piglet; TF and serum from each dead piglet were collected and tested from all study litters (Figure 1). The institutional Animal Care and Use Committee IACUC of Iowa State University, IA approved this study (IACUC-22-101).

### 2.2. Farm Eligibility Criteria

Two farms at the late stages of AASV status 1A (positive unstable at high prevalence) were identified. Selection criteria included: (a) A farrowing batch of at least 280 sows to ensure enough eligible animals; (b) agreement to not cross-foster piglets of study litters until sample collection was completed.

### 2.3. Study Herds Overview

Farm A was a 2500 breed-to-wean sow operation, managed with a four-week batch farrowing system with 280 sows each batch. The farm was PRRSV-positive stable with vaccination (AASV status II Vx) before it broke with the PRRSV 1-8-4 L1H strain. In Farm A, 106 days post LVI, TOSc samples from one whole batch of 280 sows with all parities were collected 2 weeks before their farrowing. Farm B was a continuous production system with 4500 sows. Before outbreak with a PRRSV of lineage L1C.5, the farm was naïve (AASV status IV). In Farm B, 93 days post LVI, TOSc samples from 275 gilts 2 weeks before farrowing were collected (Table 1).

### 2.4. Sample Collection

#### 2.4.1. TOSc Collection from Sows

TOSc was collected two weeks pre-farrowing and within 12 h post-farrowing without restraining the sows as previously described [13]. In brief, the rod of TOSc collector was inserted into the sow’s mouth, pointed against the tonsil area with an upwards angle, and moved back and forth for ten seconds. The qualified sample was viscous and mucous-like.

#### 2.4.2. Blood Collection from Live Piglets

Individual blood swab was collected from tail docking using disposable scalpels (Dynarex Medi-Cut Disposable Scalpels, Patriot Medical Devices, Columbus, OH, USA) and Puritan^®^ sterile polyester-tipped applicator (Puritan Medical Products Company, LLC, Guilford, ME, USA). Each swab head was placed into a 5 mL Falcon tube (Corning Science Mexico S.A. de C.V., Tamaulipas, Mexico) pre-filled with 1 mL of PBS solution.

#### 2.4.3. Tongue Fluids (TF) and Serum Collection from Dead Animals

TF was obtained from the stillborn and liveborn dead piglets as previously described with some modifications [16,17]. Briefly, for every stillborn or liveborn dead piglet from each study litter, three centimeters of tongue tissues were collected with disposable scalpels, placed in a 50 mL conical tube (Corning Science Mexico S.A. de C.V., Tamaulipas, Mexico) prefilled with 1 mL of PBS, then the tube was frozen at −20 °C and thawed at 4 °C thereafter. The resulting fluid was transferred to a 5 mL falcon tube (Corning Science Mexico S.A. de C.V., Tamaulipas, Mexico). Serum was collected using a 5-mL disposable syringe (Monoject™ Syringes, Dublin, OH, USA) to draw blood directly from heart after opening the thoracic cavity from the dead animals.

### 2.5. Diagnostic Testing and PRRSV Status Definition for Different Subpopulations Within the Litter

All samples were tested at the Iowa State University Veterinary Diagnostic Laboratory for PRRSV RNA by RT-rtPCR using validated commercially available assays. While TOSc, TF and serum samples were tested individually, blood swabs were tested in pools of seven pigs for each litter, i.e., approximately 2 tests for each litter. Ct value <40 was considered PRRSV positive.

The “live litter” PRRSV status was defined based on RT-rtPCR results of live piglets only, whereas “whole litter” PRRSV status was defined based on RT-rtPCR results of both live and dead piglets’ samples (Figure 1). Litters with at least one blood swab tested RT-rtPCR-positive were considered “live litter” positive. When there was at least one RT-rtPCR-positive result from either live or dead piglet samples, the “whole litter” status was defined as positive. Conversely, when all results from all live and dead piglet samples were RT-rtPCR-negative, the “whole-litter” was defined as negative.

When pre-farrow and/or post-farrow TOSc were tested RT-rtPCR-positive, parallel TOSc was defined as positive. Conversely, when both TOSc tested RT-rtPCR-negative, the sow’s parallel TOSc result was reported as negative.

### 2.6. Statistical Analysis

The litter was used as the unit of analysis for PRRSV RNA positivity. Specifically, if one or multiple blood swab pools, serum, or TF from dead piglets within the same litter tested positive for PRRSV RNA, the litter was considered positive for that sample type. A generalized linear mixed model with farm ID as a random variable was used to model the logit of PRRSV RNA positivity as a function of sample type on the litter level. The Tukey–Kramer test was used as a post hoc test to compare the marginal means of positivity. Dunn’s Test (non-parametric test) was performed to assess (i) differences in the Ct values from individual RT-rtPCR-positive tests between sample types and (ii) differences on the number of total born and live piglets between positive “whole litter” and its negative cohorts. All analyses were performed using the package lme4 from R program 4.2.2 (R Core Team, 2019).

Positive and negative predictive values (PPV and NPV) of TOSc and TF results for the litter status were calculated based on equations below:(1)PPV=a/(a+b)(2)NPV=d/(c+d)
where a indicates the number tested positives for both litter status and TOSc (or TF) status, b indicates the number tested positive for TOSc (or TF) status and negative for litter status, c indicates the number tested negative for TOSc (or TF) status and positive for litter status, while d indicates the number tested negative for both litter status and TOS (or TF) status.

## 3. Results

### 3.1. Number and Rate of Positive Sows and Positive “Whole Litter” and “Live Litter”

In farm A, 17 pre-farrow sows were RT-rtPCR-positive on TOSc (three were missing at farrowing) and matched with 18 negative pre-farrow sows based on parity (Table 2). From those 32 selected litters, 6 “whole litter” and 4 “live litter” were positive (positive rate was 18.8% and 12.5%, respectively). In farm B, 45 pre-farrow gilts were tested RT-rtPCR-positive (two were missing), and accompanied with 68 conveniently selected RT-rtPCR-negative gilts. From those 111 selected litters in farm B, 23 “whole” litter and 20 “live litter” were positive (positive rate was 20.7% and 18.0%, respectively). The positivity of selected sows between farm A (43.8%) and farm B (38.7%) were similar. The positivity of “whole litter” and “live litter” between two farms was similar as well. Thus, the following results were based on data merged from two farms.

### 3.2. PRRSV RT-rtPCR Positivity and Ct Values Comparison Between Samples

In general, sow TOSc had numerically higher positivity than all sample types from piglets. Statistically, pre-farrow TOSc samples had significantly higher RT-rtPCR positivity (34.3%, 95% CI, 20.6–51.1%) than blood swab pools (8.0%, 95% CI, 3.5–17.5%), dead piglet serum (4.7%, 95% CI, 1.8–11.8%) and TF (10.9%, 95% CI, 5.0–22.0%), with *p* < 0.001 for all comparisons with pre-farrow by Tukey test. Post-farrow TOSc had significantly higher positivity (21.0%, 95% CI, 11.2–35.9%) than blood swab pools and dead piglet serum (*p* = 0.014 and *p* < 0.001, respectively). The positivity difference between other groups was not significant (Tukey test, *p* > 0.05) (Table 3).

In contrast, all the sample types from piglets had numerically lower average Ct values than sow TOSc samples. Statistically (Dunn Test), dead piglet serum samples had significantly lower Ct values than TF (*p* = 0.031), blood swabs (*p* = 0.0018), pre-farrow TOSc (*p* < 0.001), and post-farrow TOSc (*p* < 0.001), while TF group had significantly lower Ct values than the pre-farrow TOSc (*p* = 0.0153) and post-farrow TOSc (*p* = 0.0015). No significant difference was observed among other groups (Figure 2).

Since the NPV and PPV were similar in both farms, predictive values of sow TOSc status for “live litter” status were based on the combined data from both farms and merged on Table 4. PPV of pre-farrow TOSc, post-farrow TOSc and parallel TOSc testing for “live litter” was 22.8% (95% CI,11.9–33.7%), 30.2% (95% CI, 16.5–44.0%), and 25.7% (95% CI, 15.5–36.0%), respectively. In all litters with dead animals, PPV of TF for “live litter” was 58.3% (95% CI, 38.6–78.1%). All TOSc and TF samples had high NPV for “live litter” status: NPV of pre-farrow TOSc, post-farrow TOSc, TOSc parallel testing, and TF for “live litter” was 87.2% (95% CI, 80.2–94.3%), 89.0% (95% CI, 82.9–95.1%), 91.7% (95% CI, 85.5–98.1%), 95.4% (95% CI, 88.8–100%), respectively.

### 3.3. Comparison of Number of Total Born and Live Piglets Between Positive “Whole Litter” and Negative “Whole Litter”

Reproductive data was retrieved from farm B. There was no significant difference in average total born between positive and negative “whole litters” (Dunn’s test, *p* = 0.65) (Table 5). However, the average number of live piglets in negative “whole litters” was significantly higher than that in positive groups groups (Dunn’s test, *p* < 0.001)

## 4. Discussion

In this study, we evaluated PRRSV RNA detection among different sub-populations within a litter with relatively easy-to-collect sow TOSc samples and litter samples including live piglet blood swabs, dead piglets TF and serum. The study also assessed predictive values of TOSc and TF for live piglets’ PRRSV status and reported reproductive data for groups of litters of each PRRSV status (positive or negative). Sows constitute a major source of PRRSV transmission to piglets [4]. Several reports have documented the presence of live PRRSV and its RNA in the cell fractions and whey of milk, identifying milk as a potential route for virus transmission to piglets [18,19]. To reduce the likelihood of virus transmission from sows to piglets and to maximize the probability of the test result reflecting the status (positive or negative) of piglets at birth and indicating PRRSV in-uterus infection, we collected piglet samples within 12 h post-farrowing as this timing aligns with reports suggesting that viremia can be detected 12 h after infection [8]. This study design can be used in further studies to investigate the occurrence of in-uterus PRRSV infection and other vertically transmitted diseases in the field. Furthermore, as PRRSV can also be transmitted in utero, as well as through milk and other secretions from sows, this underscores the importance of monitoring sow status. Utilizing TOSc, an easy and practical sample type, offers a valuable tool for effective surveillance and management of PRRSV transmission.

TOSc was collected from sows since it was reported to have comparable PRRSV positivity with tonsil scraping while being easier to collect and more animal welfare-friendly without snaring the sows [13,14,15]. This is the first report to document the PRRSV positivity and Ct values in sow TOSc samples and respective piglet samples around 90 days after deliberate exposure to live resident virus under commercial settings. Moreover, both pre- and post-farrow sow TOSc RT-rtPCR results had higher PRRSV positivity than the respective subsequent piglet sample types. Even though the role of TOSc positive sow on PRRSV transmission to piglets after birth needs further elucidation, we isolated the wildtype virus from one sow sample from Farm A, indicating possibilities of direct viral transmission from sow to piglets after they are born.

TF was collected with some modifications from dead piglets because it was reported as an easy and effective aggregate sample type for PRRSV detection [16,17]. Significantly lower Ct values in serum than TF is within expectation since TF was diluted with PBS. Numerically higher PRRSV-RNA positivity of TF compared to serum might be attributed to the hypothesis that TF might contain other fluid than blood including placental fluids, and thus may not only reflect systemic infection in the dead piglets [16]. Compared with TOSc and blood swabs from live piglets, serum and TF from dead animals had significantly lower Ct values. Specifically, dead piglet serum had median Ct value of 20.3, indicating that dead animals had a higher risk of containing a higher number of PRRSV RNA copies and potentially spreading the virus compared to live neonates. As a routine farm practice, dead animals were removed early in the day, which may increase the chances of spreading the virus. Bio-management practices such as rescheduling dead piglet disposal to the end of the day might help reduce this risk.

In this study, 2019 live piglets were collected and tested to reflect the PRRSV status of live neonates. Serum collection for such a large number of neonatal animals is difficult to implement due to their small size. Thus, blood swabs were collected individually with some modifications by swabbing blood from docked tails using disposable scalpels. It was documented that pools of blood swabs had highest sensitivity than other piglet’s sample types, including FOF, nasal swabs pools and oral swabs pools [20]. The easiness-to-handle makes blood swabs pools from tail docking a good alternative sampling method to demonstrate the PRRSV status of neonatal live piglets in field studies.

Although fetuses are susceptible to PRRSV infection at any stage of gestation upon direct intra-fetal inoculation, the virus may not cross placenta until late gestation [21,22,23]. The detection of PRRSV RNA in litter samples within 12 h post-farrowing suggests in utero infection even 90 days post-live virus exposure, while its presence in sow TOSc samples indicates a potential role of viral persistence in sows, contributing to vertical transmission. This underscores the importance of identifying and managing persistently infected sows to mitigate the risk of vertical transmission even after the visual recovery of reproductive performance after live virus exposure.

As TOSc and TF are easy to collect samples [13,16,17], we estimated the predictive values of both sample types in reflecting the live litter status. Three months after live-virus inoculation (LVI) and herd closure, the TOSc samples had an overall 25% PPV for “live litter” PRRSV status. In contrast, the TOSc samples and TF had relatively high NPV for “live litter” PRRSV status: NPV of 87.2% (95% CI, 80.2–94.3%), 89.0% (95% CI, 82.9–95.1%), 91.7% (95% CI, 85.5–98.1%) and 95.4% (95% CI, 88.8–100%) for pre-farrow TOSc, post-farrow TOSc, parallel TOSc and TF, respectively. This shows that the NPV for “live litter” status can be improved by parallel testing.

In Farm B, we successfully collected the reproductive data for all the study litters due to easy access of digital production platform. Interestingly, when we categorized the “whole litter” as positive and negative groups based on their PRRSV status, positive “whole litter” had significantly lower number of average live piglets while having similar amount of average total born compared with negative cohorts. As total born minus live piglet equals neonatal losses, the difference in reproductive parameter between PRRSV positive and negative groups lies in the dead animals. This also supports that TF from dead animals at birth are risk-based samples. An epidemiological study reported that the majority of herds recover average reproductive performance by 10 weeks post-LVI [3]. It was also reported that during PRRSV transition stages, from 10 weeks after PRRSV outbreak until PRRSV low prevalence (AASV status 1B), there was a significant increase of one liveborn [24]. However, our data showed that 13 weeks after LVI, positive “whole litter” had 4.5 fewer live piglets/litter compared to negative litters, highlighting that under the conditions of this study, PRRSV detection in PRRSV-unstable at low prevalence farm was an important risk factor for pig livability. This also highlights the importance of developing methods such as test removal or test segregation to get rid of the positive litters, which not only has inferior reproductive parameter but also higher risk of spreading the virus.

Higher probability of PRRSV detection of TOSc in sows, low Ct values of serum and TF from dead piglets, inferior reproduction performance in positive litters, and high NPV of TOSc and TF highlight the opportunities of test segregation of TOSc positive sows and upgrade of bio-management measures to handle dead piglets for reducing vertical transmission and reaching a sooner PRRSV stability.

There are some limitations in this field study. First, we only had access to complete reproductive data records in Farm B. Second, some of the mummies were too small to sample, so no mummies were sampled in this study, and PRRSV RNA detection in this subpopulation was not evaluated. Thirdly, it is also possible that live piglets blood swab samples were contaminated by positive dead piglet samples, including mummies, because of frequent contact between live and dead piglets within the crate. These can affect the predictive values of both sample types. For example, in six parallel TOSc negative while “live litter” positive litters, 3 litters were having 2, 7, 9 mummies, respectively, while another 2 litters having Ct value of 39.2 and 39.3, respectively.

## 5. Conclusions

In conclusion, TOSc showed higher PRRSV RNA positivity than piglet sample types while dead piglet serum and TF had significantly lower Ct values than TOSc. Moreover, TOSc samples had low PPV and high NPV for their respective litter’s PRRSV status. Likewise, TF was a good predictor of neonatal live piglet PRRSV status.

## Figures and Tables

**Figure 1 vetsci-12-00150-f001:**
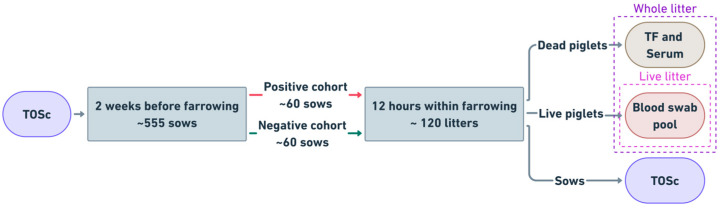
Schematic illustration of study design. TOSc, tonsil oral scrubbing; TF, tongue fluid. The “live litter” is defined as litter taking only live piglets into account. The “whole litter” is defined as litter comprising both live and dead piglets. RT-rtPCR Ct value < 40 was deemed as positive.

**Figure 2 vetsci-12-00150-f002:**
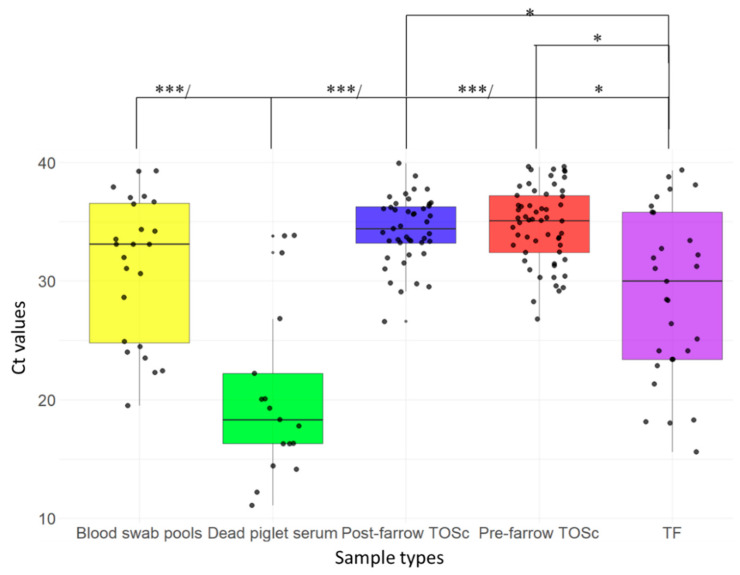
Ct values distribution of positive samples from different sample types. TF tongue fluid. Dead piglet serum, serum from dead animals. TOSc, tonsil oral scrubbing. * indicated *p* < 0.05, *** indicated *p* < 0.001 (Dunn’s test for pairwise comparison).

**Table 1 vetsci-12-00150-t001:** Overview of study farms.

Farm	PRRSV RFLP and Lineage	Herd Type	Sow Inventory	Production Type	AASV PRRSV Status Before Outbreak	Screened Population	Screening Timepoint (2 Weeks Before Farrow)	Reproductive Data Collected
A	1-8-4, 1H	Breed-to-wean	2500	4-week batch	II-Vx	280 pregnant sows with parities 1–9	106 days after LVI	No
B	1-4-4, L1C.5	Breed-to-wean	4500	Continuous	IV	275 pregnant gilts	93 days after LVI	Yes

**Table 2 vetsci-12-00150-t002:** Number of positive and negative sows and “whole litter” and “live litter” in each farm.

Farm	Number of Positive Sows Identified/Total Number of Sows Screened	Number and Rate of Positive Sows Selected
A	17/280	14 (43.8%)
B	45/275	43 (38.7%)

**Table 3 vetsci-12-00150-t003:** PRRSV RNA positivity and Ct range of RT-rtPCR-positive samples among sample types.

	Pre-Farrow TOSc	Post-Farrow TOSc	Blood Swab Pools	TF	DEAD Piglet Serum
PRRSV RNA positivity (95% CI)	34.3% (20.6–51.1%) ^a^	21.0% (11.2–35.9%) ^ab^	8.0% (3.5–17.5%) ^c^	10.9% (5.0–22.0%) ^bc^	4.7% (1.8–11.8%) ^c^
Ct value mean (range)	34.7 (26–39.6)	34.2 (26.6–39.9)	31.2 (19.5–39.3)	28.9 (15.6–39.3)	20.3 (11.1–33.8)

TF, tongue fluid. Dead piglet serum, serum from dead animals. TOSc, tonsil oral scrubbing. ^a^, ^b^, ^c^, Different letters indicate significant differences in least square means (Tukey test, *p* < 0.05). *p* = 0.76 for pairwise comparison between dead serum and blood swab pools, *p* = 0.31 for pairwise comparison between dead serum and TF, *p* = 0.41 for pairwise comparison between pre-farrow TOSc and Post-farrow TOSc, *p* = 0.32 for pairwise comparison between Post-farrow TOSc and TF, *p* = 0.94 for pairwise comparison between blood swab pools and TF.

**Table 4 vetsci-12-00150-t004:** Summary table for predictive values of TOScs samples and TF samples PRRSV status for live litter PRRSV status.

	Pre-Farrow TOSc PRRSV Status	Post-Farrow TOSc PRRSV Status	Parallel TOSc PRRSV Status	TF PRRSV Status
NPV with 95% CI	87.2% (80.2–94.3%)	89.0% (82.9–95.1%)	91.7% (85.5–98.1%)	95.4% (88.8–100%)
PPV with 95% CI	22.8% (11.9–33.7%)	30.2% (16.5–44.0%)	25.7% (15.5–36.0%)	58.3% (38.6–78.1%)

Pre-farrow TOSc, tonsil oral scrubbing from gestation sows; post-farrow TOSc, tonsil oral scrubbing from sows 12 h within farrowing; TF, tongue fluid. NPV, negative predictive value, PPV, positive predictive value. 95% CI, 95% confidence interval.

**Table 5 vetsci-12-00150-t005:** Comparison of average total born and average live piglets between positive “whole litter” and negative “whole litter”.

	Average Number of Total Born with 95% CI	Average Number of Live Piglets with 95% CI
Positive “whole litter”	13.9 (11.3–16.5) ^a^	9.0 (7.3–11.0) ^a^
Negative “whole litter”	14.8 (14.1–15.6) ^a^	13.5(12.8–14.2) ^b^

^a^/^b^: Different superscript letters indicate significant differences in means by Non-parametric Dunn’s test (*p* < 0.05). The “whole litter” is defined as litter comprising both live and dead piglets. If any TFs or blood swabs from one litter were RT-rtPCR-positive, the “whole litter” was deemed PRRSV-positive.

## Data Availability

Data is contained within the article.

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
