# Peer review of "PRRSV RNA Detection and Predictive Values Between Different Sow and Neonatal Litter Sample Types"

_vetsci, 2025, doi:10.3390/vetsci12020150_

Round 1

Reviewer 1 Report

Comments and Suggestions for Authors

This study investigated the RNA detection of porcine reproductive and respiratory syndrome virus (PRRSV) in sows and newborn piglets and its predictive value, in particular to determine the potential for vertical transmission through different sample types. This type of research was of important clinical meaning. It provide a new method to differentiate vertical and horizontal transmission. I agree to accept the article after minor revision. It need to be discussed whether PRRSV can be transmitted by milk.

Author Response

Comment 1: 

This study investigated the RNA detection of porcine reproductive and respiratory syndrome virus (PRRSV) in sows and newborn piglets and its predictive value, in particular to determine the potential for vertical transmission through different sample types. This type of research was of important clinical meaning. It provides a new method to differentiate vertical and horizontal transmission. I agree to accept the article after minor revision. It need to be discussed whether PRRSV can be transmitted by milk.

Response 1: Thanks for the acknowledgement to the manuscript and the constructive comments. Whether PRRSV can be transmitted by milk is of great importance both in the field and to our study design. Several reports have documented the presence of live PRRSV and its RNA in the cell fractions and whey of milk, identifying milk as a potential route for virus transmission to piglets. To reduce the likelihood of virus transmission from sows to piglets after birth—whether through milk or direct contact—we collected piglet samples within 12 hours post-farrowing. This timing aligns with reports suggesting that viremia can be detected 12 hours after infection. Furthermore, as PRRSV can also be transmitted in utero, as well as through milk and other secretions from sows, this underscores the importance of monitoring sow status. Utilizing TOSc, an easy and practical sample type, offers a valuable tool for effective surveillance and management of PRRSV transmission. Discussion is added as requested in lines “264-281”

Reviewer 2 Report

Comments and Suggestions for Authors

In this study, Li et al. characterized the detection of PRRSV RNA among all subpopulations within a litter using easy-to-handle TOSc samples and neonatal litter samples, suggesting that vertical transmission occurs 90 days post-LVI in sows. The TOSc samples exhibited a higher positivity rate for PRRSV RNA compared to piglet sample types, while serum from deceased piglets and TF demonstrated significantly lower Ct values than TOSc. Overall, the research design is reasonable, and the data is credible; however, some issues need to be addressed.

1. The abstract section needs to provide an introduction to PRRSV.

2. Suggest adding a table to introduce information about the samples.

Author Response

Comment 1: The abstract section needs to provide an introduction to PRRSV.

Response 1: Thanks for the comment. Introduction to PRRSV was added in the abstract section as requested. See lines “29-30”.

Comment 2: Suggest adding a table to introduce information about the samples.

Response 2: Thanks for the comments. The detailed information of samples collected were illustrated in Figure 1 in the manuscript. A total of 555 TOSc samples were collected from pre-farrowing sows. Within 12 hours post farrowing, TF and serum were collected from dead piglets, blood swabs collected from live piglets, and TOSc collected again for 147 sows.
